# Exploration of Topic Classification in the Tourism Field with Text Mining Technology—A Case Study of the Academic Journal Papers

**I-Cheng Chang** [1], **Jeou-Shyan Horng** [2], **Chih-Hsing Liu** [3], **Sheng-Fang Chou** [4] **and Tai-Yi Yu** [5,*]

1. Department of Environmental Engineering, National Ilan University, Yilan 260007, Taiwan; icchang@niu.edu.tw
2. Department of Food and Beverage, Shih Chien University, Taipei 104336, Taiwan; t10004@ntnu.edu.tw
3. Department of Tourism Management, National Kaohsiung University of Science and Technology, Kaohsiung 811532, Taiwan; phd20110909@gmail.com
4. Department of Hospitality Management, Ming Chuan University, Taoyuan 333321, Taiwan; sfchou@mail.mcu.edu.tw
5. Department of Risk Management and Insurance, Ming Chuan University, Taipei 111005, Taiwan
* Correspondence: yti@mail.mcu.edu.tw

**Abstract:** This study collects abstracts of SSCI tourism journal papers between 2010 and 2019 from the WoS (Web of Science) database and uses a novel method of topic classification to explore the vocabulary characteristics of the classified articles. The corpora of abstracts are given quantitative Term Frequency–Inverse Document Frequency (TF–IDF) weights. A hierarchical K-means cluster analysis is then performed to automatically classify the articles; co-word analysis techniques are used to show the characteristics of feature words for distinct clusters, titles, and the consistency of the classified articles. Based on the results for 5783 abstracts, cluster analysis classifies the number of K-means clusters into six categories: travel, culture, sustainability, model, behavior, and hotel. A cross-check method is applied to assess the consistency of the topic classifications, list titles and keywords of the documents with the three smallest distances in each category and apply a strategic diagram to present the features of the distinct categories.

**Keywords:** cluster analysis; text mining; word cloud; co-word analysis; strategic diagram

## 1. Introduction

In recent years, smart technology, including artificial intelligence, big data, and the sharing economy, has become an important trend leading to the development of the global smart industry. In particular, big data and artificial intelligence have become dominant in various industries, especially knowledge-intensive ones such as tourism [1]. In the era of big data, firms use artificial intelligence to analyze the huge amounts of messy data they capture to identify useful knowledge that can help them innovate business models and value propositions. By utilizing big data analysis and artificial intelligence, the tourism and catering industry can provide real-time feedback, as well as improved transparency, market segmentation, decision-making, and product and service innovation, among other aspects [2], and thereby increase the value of the industry.

Tourism is generally defined as persons traveling to and staying in places outside their usual environment, for not more than one consecutive year, for leisure, business, or other purposes [3]. The 2030 Agenda for Sustainable Development SDG target 8.9 states the following target to achieve by 2030: "devise and implement policies to promote sustainable tourism that creates jobs and promotes local culture and products" [4]. The connotation of sustainable tourism is frequently enriched: initial attention focused on environmental issues, while more definitions denote the importance of working towards the balanced development of economic, social, and environmental aspects. Most people believe that

sustainable tourism emphasizes the connection between tourism activities and society with respect to the long-term coordinated development of the economy, resources, and the environment [5]. Goals are aimed at both economic development and a reduction in the negative impact of tourism activities, which includes continued development of the tourism industry while protecting natural and cultural resources. It is vital to coordinate and balance the relationships between different stakeholders in the process of tourism development [6].

Information technology is part of the lifeblood of the tourism industry [7]. Combining knowledge gathered through statistics and domain experts from the tourism industry can help verify the results of visualization analysis. Information technology can apply an automatic topic classification to natural language processing documents to classify representative documents quickly and objectively; co-word analysis and association rule analysis can then be used to analyze the importance and relevance of specific words. There are four main research aims for this article: (1) carry out the subject classification process of academic articles in the tourism field to assess the consistency and characteristics of the topic classification; (2) assess the characteristics of the subject classification and confirm its consistency; (3) use co-word analysis and strategic diagram to understand the importance and relevance of specific marketing strategy vocabulary; and (4) recognize the research tendencies of distinct topics in tourism field.

### 1.1. Topic Classification in the Tourism Industry

In the past, the process of classifying, deconstructing, and analyzing relevant documents in this area required significant time and resources from domain experts. As this body of work continues to grow and diversify, text mining technology can better comprehend and promote the leisure industry, such that the general public becomes willing to understand, recognize, support, and participate in achieving the goals of sustainable development. Text mining can provide valuable knowledge from a large number of unstructured texts. Early text mining techniques were used in file classification [8]. As the various types of text information keep increasing, including e-books, web pages, online news pages, blog articles, images, sounds, and videos, manual capture becomes impossible, and the need for topic models for automatic classification becomes apparent. Regarding the application of text mining in the tourism field, Okumus et al. [9] investigated the catering and tourism industry from 1976 to 2016: they analyzed the evolution of food and gastronomy research and identified emerging research topics, methods, and areas of national or interdisciplinary cooperation. Most of the 462 articles centered on gourmet, quantitative, and practical topics. Sainaghi et al. [10] used a cross-reference network analysis to evaluate the literature on hotel performance published between 1996 and 2015 to identify the most cross-cited papers, authors, and journals. Their sample analysis included 734 papers and demonstrated a spectacular growth of outputs, with the last time period (2011–2015) contributing 56% of output; in total, 1% of the sample accounted for 14% of the cross-references.

The topic model can use text mining algorithms, keyword libraries, and keyword occurrence ratios from a large amount of unstructured text data to define the subjective or objective category of the documents. Topic classification algorithms commonly used in text mining include cluster analysis, logistic regression, boost tree regression, hierarchical K-Means, K-means, latent Dirichlet allocation (LDA), and support vector machine (SVM) [11,12]. The hierarchical K-Means is very commonly applied in tourism management, mainly for classification issues, such as the attribute classification of tourists [13–15] and motivation classification of tourists [16,17]. With the hierarchical K-means cluster analysis, Suni and Komppula [16] used 30 motivational statements to classify respondents into one of five groups: controllers, indifferent, nostalgia, comfort seekers, and novelty seekers. Lee and Kim [14] divided the older adult volunteers in an international sporting event into two distinct segments of serious leisure characteristics, while Michèle et al. [15] analyzed the activity profiles of social and leisure activities among older adults and divided the respondents into seven clusters. Finally, Jiao et al. [17] classified cruise ship tourists into four main categories: psychometric tourists, traditional tourists, pioneers, and sightseers.

With the help of the LDA model, Jia [18] randomly selected 100 yoga centers in Shanghai, identified 15 topics with the top 10 words from review comments. Vu et al. [19] utilized topic modeling to perform a travel itinerary analysis using the LDA model to clarify information on itineraries and tourist preferences. The optimal number of topics was decided as 24 using validation perplexity and computation times for different topic numbers with a total of 12,446 daily itineraries; each topic was visualized using the word cloud method and a heat map diagram. Shafqat and Byun [20] applied the LDA model to a travel blog database, extracted the top 150 blogs on tourism in Jeju, Korea, and identified the top 11 topics: location, timing, food, weather, entertainment, environment, accommodations, transportation, expense, services, and rental cars. Sutherland and Kiatkawsin [21] applied the LDA approach to identify 43 topics of interest that drive customer experience and satisfaction within a dataset of 1,086,800 Airbnb reviews; they grouped them into four topics: evaluation, location, unit, and management characteristics. The number of suitable classifications and sound interpretations of the topic categories are important for topic modeling.

Pleumarom [22] studied the tourism industry in the Mekong region and stated that the local government must adopt a cohesive management approach to achieve sustainable tourism in areas where multiculturism and government institutions coexist. Sustainable tourism is an important topic in the tourism and hospitality industries; it can improve organizational performance, help gain competitive advantage, and be used as a commercial marketing topic. There are many research themes and influencing factors of sustainable tourism, such as sense of place, pro-environmental behaviors [23–25], and human health [26]. The interpretation ability of the local tourism industry could increase the income of sustainable tourism, and local interpreters could meet customer needs and create local employment, promote economic sustainability, and also act as on-site supervisors of visitors to influence their understanding of local perspectives, social protection, and environmental issues [27,28].

### 1.2. Marketing Strategy in the Tourism Industry

Marketing strategy plays a very important role in tourism. The traditional 4P marketing strategy proposed by McCarthy [29], focused on promotion, place, product, and price, has been widely applied in various business fields. Booms and Bitner [30] revised McCarthy's [29] 4P to 7P, adding people, physical evidence, and process. Kolter [31] revised McCarthy's [29] 4P to 6P, adding politics and public opinion to provide marketing strategies for a complex and diverse society. Pomering, Noble, and Johnson [32] applied 10 marketing foundations to the marketing model of sustainable tourism: promotion, place, product, price, physical evidence, process, packaging, participation, programming, and partnership while considering economic, environmental, and social issues. Dudensing et al. [33] indicated that the different marketing strategies of stakeholders in the tourism industry led to heterogeneity or conflict in terms of stakeholder expectations. Wray [34] emphasized the interactive nature of sustainable tourism—ensuring that profits remain with local operators and sites. Lozano-Oyola et al. [35] also supported the aforementioned view, advocating that those stakeholders review economic indicators before making sustainable tourism decisions. Different expectations of various stakeholder groups may cause conflicts: destination marketing must take into account the views of numerous stakeholders [36]. Generally, marketing and sustainability can work together through the development of a sustainable tourism marketing model, such as managing the travel route of the tourism industry through an ecological footprint.

Big data plays a catalytic role in the process of determining consumer preferences. By obtaining correct data, meaningful analysis can take place, leading to structural changes in consumer behavior models and marketing strategies [37]. Through appropriately identified disseminations, Samara et al. [38] noted that the benefits of adopting big data and artificial intelligence strategies include increased efficiency, productivity, and profitability for tourism suppliers, combined with an extremely rich and personalized experience for

travelers. Katsikari et al. [39] proposed that the rapid expansion of the Internet and social media provides marketers with simple and cost-effective ways and opportunities to reach potential tourists; their study investigated which destination elements are deemed attractive by tourists who use social media.

Through the text mining results of a large number of academic articles, important words related to tourism marketing can be obtained and, at the same time, their relevance and relative importance can be understood. Based on the existing information technology, the tourism industry can expand consumerism-based IT and tourism research in order to participate in a wider dialogue; emphasize the use of technology to achieve a better quality of life, economic prosperity, social well-being, and sustainability; and use open data and shared social knowledge as a basis for tourism experience and innovation [40].

### 1.3. Co-Word Analysis and Strategic Diagram

Co-word Analysis is a quantitative technique for scanning document content to denote when and where a defined specific word co-occurs. Through co-word analysis, various co-occurrence relationships in a specific object can be expressed, such as co-quotation, co-author, and co-word characteristics [41]. A strategic diagram is developed from the analysis, which assists in identifying evolutionary trends and relationships between thematic groups [42].

In the application of co-word analysis, Guo et al. [43] surveyed 1138 articles and reviews from 1980 to 2016 and used 52 high-frequency keywords related to company restrictions to investigate the current situation and trends of company restrictions. The central terms were "restrictions", "learning", "institutions", and "behavior"; the results show that "restrictions" had the highest degree of importance. The aforementioned 52 high-frequency keywords could be divided into six categories, and the indicators of company development (such as innovation, supply chain, decision-making, performance, sustainability, and employee behavior) were significantly related to company restrictions. Khasseh et al. [44] used co-word analysis to describe the topic characteristics within two journals, *Scientometrics* and *Journal of Informetrics*, from 1978 to 2014; they then divided them into 11 representative topics using hierarchical cluster analysis and utilized a strategy diagram to illustrate the structure, maturity, and cohesion of each topic. Corrales-Garay et al. [45] applied co-word analysis to create a map of the main themes identified in the knowledge areas and determined their importance and relevance.

Leung et al. [46] sampled 406 publications related to social media from 2007 to 2016 across 16 business and hospitality/tourism journals and applied co-word analysis to identify the evolution of research themes over time. Shen et al. [47] collected 29 years of online database data of academic journals and then used co-word analysis and bibliographic analysis techniques to analyze trends, core authors, degrees of cooperation, core journal analysis, and distribution of publishing institutions. This allowed them to establish 10 important and unevenly distributed research trends and then propose a new potential research theme of information search and information security. De la Hoz-Correa et al. [48] utilized co-word analysis to denote six clusters of themes in published research listed in the Web of Science (WoS) and Scopus database; this type of analysis offers powerful insights into the conceptual structure of medical tourism research. The co-word analysis is an effective manner to identify the content, importance, and relevance of different themes from the aforementioned references.

Rodríguez-López et al. [49] applied a strategic diagram to present the importance of topic themes from a bibliometric analysis of published academic research dealing with restaurants in the fields of hospitality, leisure, sport, and tourism ring the period from 2000 to 2018. Muñoz-Leiva et al. [50] conducted data mining on 759 papers related to blockchain technology in the financial field by employing co-word analysis and strategic diagrams to explore hot topics and predict future development trends. Rodríguez-López et al. [51] selected documents whose titles included specific terms from two online databases, Web of Science (WoS) and Scopus, and utilized a keyword strategic diagram to determine the

importance of keywords and their levels of development. Terán-Yépez et al. [52] sampled 216 articles from sustainable entrepreneurship and identified the most significant research tendencies, enabling the proposal of several future research directions through graphic mapping of strategic diagrams. Finally, Jiménez-García et al. [53] applied bibliometric techniques to investigate research trends in 214 articles related to sports tourism and sustainability and used strategic diagrams to identify the most significant research tendencies across distinct topics.

## 2. Materials and Methods

### 2.1. Selection and Word Segmentation of Corpus

Based on the limitations of research timeliness, complexity, and professional level, the research topic is the tourism-related field of SSCI journals, and the subject field is hospitality, leisure, sport, and tourism. There are more than 20 highly relevant SSCI journals in this field based on their subject, timeline (2010–2019), richness, and representativeness. This paper initially selected 11 journals from the WoS database that fully complied with the objectives of this paper, and then reduced that total to 8 by retaining only those with leisure or tourism in their title. The corpus was then set as the abstracts for the papers that appeared in these journals, with text preprocessing performed with the NLTK package in Python, which included text extraction, word segmentation, and deletion of stop words, numbers, spaces, English letters, and part-of-speech tagging, among others.

There are two important steps to achieve topic classification: determining the importance of vocabulary and choosing a topic model. The text mining technique often uses TF or TF–IDF weights to determine the importance of vocabulary. Based on many previous research recommendations, this study uses TF–IDF weights [54–56]. The NLP algorithm performs word segmentation to convert the corpus into a "structured" data style. Based on suggestions from the literature [57], this study utilized the Jieba word segmentation algorithm on the Python platform. The detailed five steps for extracting feature words were shown as (1) preprocessed text cleaning for abstracts of journals papers; (2) word segmentation with Jieba; (3) filter tokens, filter stopped words, perform other cleaning rules, and replace tokens; (4) construct and confirm feature wordlist with domain experts, and (5) calculate TF–IDF weights. In addition, feature keywords identified in the word segmentation must appear in at least 6% of the abstracts, and the structured DTM data are determined by all of the academic paper abstracts. Three domain experts with domain knowledge in the tourism field were invited to extract, confirm, and determine feature keywords, calculate TF–IDF weights, build the DTM matrix, perform both topic classification and co-word analysis, verify text mining results, and extract the required information, as well as verify the consistency of the topic classification and corpus data. The TF–IDF weight includes text frequency (TF) and inverse document frequency [58,59]. The weighted score of a single word in each article can be calculated by the two indicators of TF and IDF as in the formula:

$$TF - IDF_{ij} = TF_{ij} \times IDF_i \qquad (1)$$

$TF_{ij}$: shows the text frequency that keyword $t_j$ occurs in document $d_i$.

$IDF_i$: shows the inverse value of document frequency ($df_i$), where $df_i$ is the document frequency of keyword $t_i$. Sometimes, the $IDF_i$ factor has many variants, such as $\log (N/df_i + 1)$, $\log(N/df_i) + 1$, and so on. The TF–IDF weights matrix can be constructed based on feature keywords and documents.

### 2.2. Hierarchical K-means Cluster Analysis

To perform topic classification, this study utilized the hierarchical K-means analysis method: a two-stage and commonly utilized method for clustering. The first stage uses a hierarchical cluster analysis to determine the number of K clusters, and the second stage uses the K-means analysis to divide the documents into K clusters based on the hierarchical cluster analysis. For the hierarchical cluster analysis method, this study used the Euclidean distance to define its similarity and the Wards' Method (minimum variance)

linkage method. The appropriate number for topic classification was decided through discussions with the tourism industry domain experts and residual values. The K-means cluster analysis method involved four steps: (1) select the center points of the K clusters in order; (2) calculate the distance from each sampling to the cluster center and assign the sampling points to the closest cluster; (3) redistribute the sample points to the K-th cluster; and, (4) if the redistributed samples met the adjustment rule condition, steps (2) and (3) were repeated until the convergence condition was reached. The hierarchical clustering and K-means clustering processes were conducted using IBM SPSS software.

### 2.3. Co-Word Analysis

This study adhered to the co-word analysis procedure recommended by Ding et al. [41] to better understand academic article evolution and trends across various tourism topics: (1) build a co-word binary matrix based on co-occurrences of feature words in each article; (2) utilize co-word analysis to analyze the relevance of sales keywords based on topic classification from the hierarchical K-means and keywords of marketing strategies; (3) build web diagrams based on the binary co-word analysis of marketing keywords to understand the importance, relevance, and visual analysis results of marketing keywords; and (4) construct a strategic diagram to recognize the characteristics of each topic category. A strategic diagram [49,50] is a two-dimensional graph where the horizontal axis and the vertical axis represent the centrality and density, respectively, and the origin represents the average value of the centrality and density. The density reveals the connection strength between each word and other words in the same category, and centrality shows the connection strength between individual keywords and other keywords within other clusters.

### 3. Results and Discussion

### 3.1. Corpus Structure

Based on the number of articles published in different journals each year (Table 1), this study selected 5783 effective abstracts during the period 2010–2019. The number of papers displayed an increasing trend: there were 377 in 2010 and 816 in 2019. The least frequent topic across the 10-year period was leisure sciences (314, 5.4%), with the smallest number (25 papers) appearing in years 2011 and 2015, and the largest number (48 papers) appearing in 2018. The most frequent topic was tourism management (1779, 30.8%), for which the smallest and largest number of yearly papers appeared in 2010 (95 papers) and 2017 (240 papers), respectively.

**Table 1.** Numbers of articles for distinct journals in 2010–2019.

| Source Title | 2010 | 2011 | 2012 | 2013 | 2014 | 2015 | 2016 | 2017 | 2018 | 2019 | Total |
|---|---|---|---|---|---|---|---|---|---|---|---|
| Annals of tourism research | 52 | 71 | 90 | 75 | 76 | 54 | 62 | 72 | 57 | 124 | 733 |
| Current issues in tourism | 32 | 45 | 53 | 45 | 64 | 70 | 85 | 111 | 122 | 162 | 789 |
| Journal of hospitality & tourism research | 26 | 24 | 24 | 24 | 26 | 22 | 32 | 41 | 57 | 61 | 337 |
| Journal of sustainable tourism | 61 | 44 | 60 | 64 | 63 | 74 | 90 | 107 | 117 | 96 | 776 |
| Journal of travel & tourism marketing | 56 | 53 | 52 | 54 | 62 | 75 | 86 | 85 | 91 | 75 | 689 |
| Leisure sciences | 30 | 25 | 30 | 31 | 29 | 25 | 28 | 36 | 48 | 32 | 314 |
| Tourism geographies | 25 | 27 | 28 | 33 | 56 | 42 | 31 | 45 | 39 | 40 | 366 |
| Tourism management | 95 | 155 | 160 | 154 | 145 | 196 | 194 | 240 | 214 | 226 | 1779 |
| Sum | 377 | 444 | 497 | 480 | 521 | 558 | 608 | 737 | 745 | 816 | 5783 |

The top 30 feature words from the abstracts in terms of text frequency and document frequency are listed in Table 2. The text frequency numbers for the top 10 feature terms were *tourist* (5246), *destination* (3704), *model* (2549), *experience* (2426), *development* (2346), *social* (2238), *travel* (2127), *behavior* (2022), *relationship* (1863), and *hotel* (1772). The document frequency numbers for the top 10 feature terms were *tourist* (2068), *destination* (1524), *model* (1484), *development* (1284), *relationship* (1247), experience (1220), *implication* (1218), *social* (1216), *impact* (1125), and *effect* (1074).

**Table 2.** The top 30 terms with high term frequency and document frequency.

| Rank | Terms | Term Frequency | Terms | Document Frequency |
|---|---|---|---|---|
| 1 | tourist | 5246 | tourist | 2068 |
| 2 | destination | 3704 | destination | 1524 |
| 3 | model | 2549 | model | 1484 |
| 4 | experience | 2426 | development | 1284 |
| 5 | development | 2346 | relationship | 1247 |
| 6 | social | 2238 | experience | 1220 |
| 7 | travel | 2127 | implication | 1218 |
| 8 | behavior | 2022 | social | 1216 |
| 9 | relationship | 1863 | impact | 1125 |
| 10 | hotel | 1772 | effect | 1074 |
| 11 | impact | 1759 | influence | 1012 |
| 12 | effect | 1736 | behavior | 1011 |
| 13 | community | 1659 | role | 979 |
| 14 | economy | 1642 | economy | 961 |
| 15 | visitor | 1604 | travel | 908 |
| 16 | culture | 1497 | management | 867 |
| 17 | service | 1430 | industry | 843 |
| 18 | influence | 1360 | culture | 833 |
| 19 | value | 1342 | theory | 818 |
| 20 | satisfaction | 1309 | strategy | 801 |
| 21 | management | 1307 | understanding | 798 |
| 22 | change | 1301 | process | 776 |
| 23 | implication | 1287 | survey | 766 |
| 24 | role | 1268 | empirical | 747 |
| 25 | local | 1235 | future | 743 |
| 26 | industry | 1228 | activity | 733 |
| 27 | intention | 1222 | local | 719 |
| 28 | activity | 1199 | nature | 717 |
| 29 | strategy | 1189 | community | 714 |
| 30 | nature | 1175 | service | 703 |

To better understand the topic classification of some of the academic tourism journals utilized, their mission statements were reviewed. "Tourism Management" is a leading international journal for all those concerned with management (including planning) of travel and tourism; "Annals of Tourism Research" is a social sciences journal focusing upon the academic perspectives of tourism, striving for a balance between theory and application, and ultimately devotes itself to the development of theoretical structures; "Current Issues in Tourism" encourages in-depth discussion and criticism of key questions within the subject, including application and theoretical work that addresses tourism inquiry, method, and practice; "The Journal of Sustainable Tourism" advances critical understanding of the relationships between tourism and sustainable development and publishes theoretical, conceptual, and empirical research that explores one or more of the economic, social, cultural, political, organizational, or environmental aspects of the subject. The text frequency and document frequency information in Table 2 reveals the importance of feature words, but not the relevance between the feature words and marketing strategies. Topic classifications may help to provide better explanations and interpretations for the relationships between feature words and marketing strategies.

To better understand trends pertaining to feature words from 2010 to 2019, this study computed the document frequency of feature words and plotted them on a contour map (Figure 1). The feature words *tourist* and *destination* ranked first or second from 2010 to 2019; *community* and *culture* showed increasing trends from 2016 on; *experience*, *model*, and *relationship* maintained their rankings across each year; *service* suddenly increased from 2015 to 2017, but then decreased in 2018; and *economy* and *social* showed increasing trends from 2014 on. The contour map of document frequency for feature words demonstrates that

individual feature words had unique time trends and did not maintain consistent trends with other feature words; analytical results show the uniqueness of the feature words.

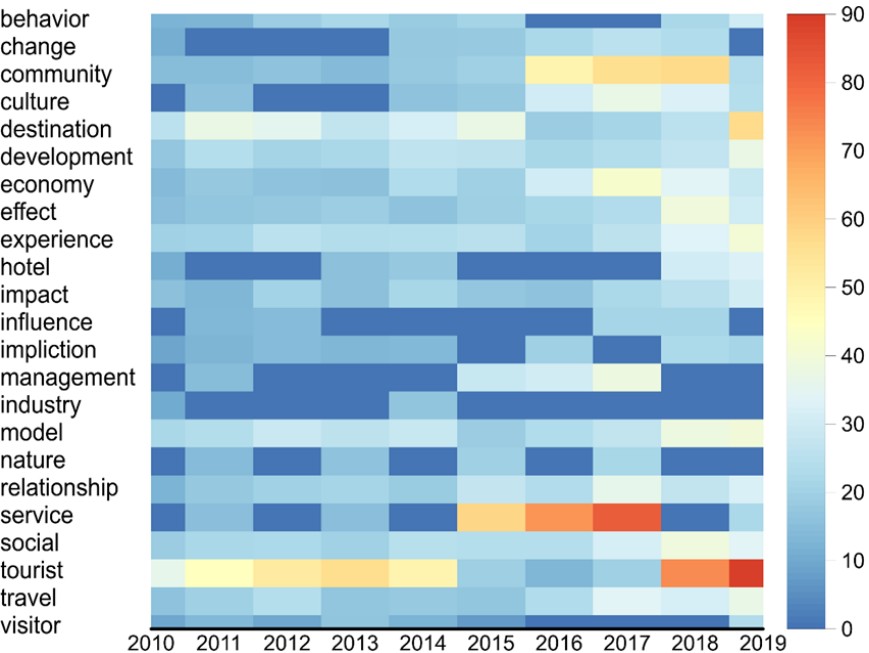

**Figure 1.** Contour diagram for number of documents of feature words during 2010–2019.

### 3.2. Topic Classification

Topic interpretability is usually determined by holistic considerations. That is, domain experts can find the best number of topic classifications that describe or explain the content of the theme by looking at the popular feature words that define the potential topic; the title or abstract of a specific document may prove beneficial. If there are too many topic categories, it becomes difficult to explain the topic category results and assess the consistency of the content and the topic. The relationship between the decreasing ratios of the total residual and the number of clusters (Figure 2) demonstrated that when the number of clusters was 5, 6, and 7, the decreasing ratios in the total residuals were 0.950, 0.802, and 0.786%, respectively. The average explained ratio of each document was 0.0173% for 5783 documents. With the increase in the number of clusters from six to seven, the increase in the explained ratio was less than 0.0155%, and the decreasing slope of the explained ratio was low. On the basis of options of domain experts and the relationship between number of clusters and decreasing ratios of residual (Figure 2), the optimum number of topics was then decided as six. The six (Table 3)topics were termed as travel (S1), culture (S2), sustainability (S3), model (S4), behavior (S5), and hotel (S6).

**Table 3.** The feature words clustered with K-means clustering method.

| Clusters | Terms with High TF–IDF Weights in Descending Order | Count and Ratios of Documents | Topic |
|---|---|---|---|
| S1 | travel, destination, tourist, experience, behavior, model, social, relationship, market, online, motivation, traveler, decision, change, group, intention | 389 (6.73%) | Travel |
| S2 | tourist, destination, experience, behavior, culture, model, image, travel, site, social, satisfaction, effect, relationship, Impact, visitor, motivation, information | 927 (16.0%) | Culture |

**Table 3.** *Cont.*

| Clusters | Terms with High TF–IDF Weights in Descending Order | Count and Ratios of Documents | Topic |
|---|---|---|---|
| S3 | development, social, community, leisure, culture, experience, local, sustainable, change, nature, activity, process, resident, impact, economy | 2153 (37.3%) | Sustainability |
| S4 | model, economy, impact, effect, country, demand, policy, relationship, development, tourist, destination, industry, market, variable | 666 (11.5%) | Model |
| S5 | behavior, service, intention, satisfaction, visitor, customer, experience, relationship, effect, model, value, quality, influence, implication, tourist | 828 (14.3%) | Behavior |
| S6 | destination, hotel, image, management, performance, effect, model, tourist, marketing, strategy, industry, implication, review, service, brand, relationship, market | 820 (14.2%) | Hotel |

The terms related to 7P that could be described as follows: promotion (satisfaction, experience, nature), place (destination, local, hotel), product (hotel, destination, nature, activity), people (tourist, service, visitor), physical evidence (hotel, nature), process (model, activity), and price (economy, value).

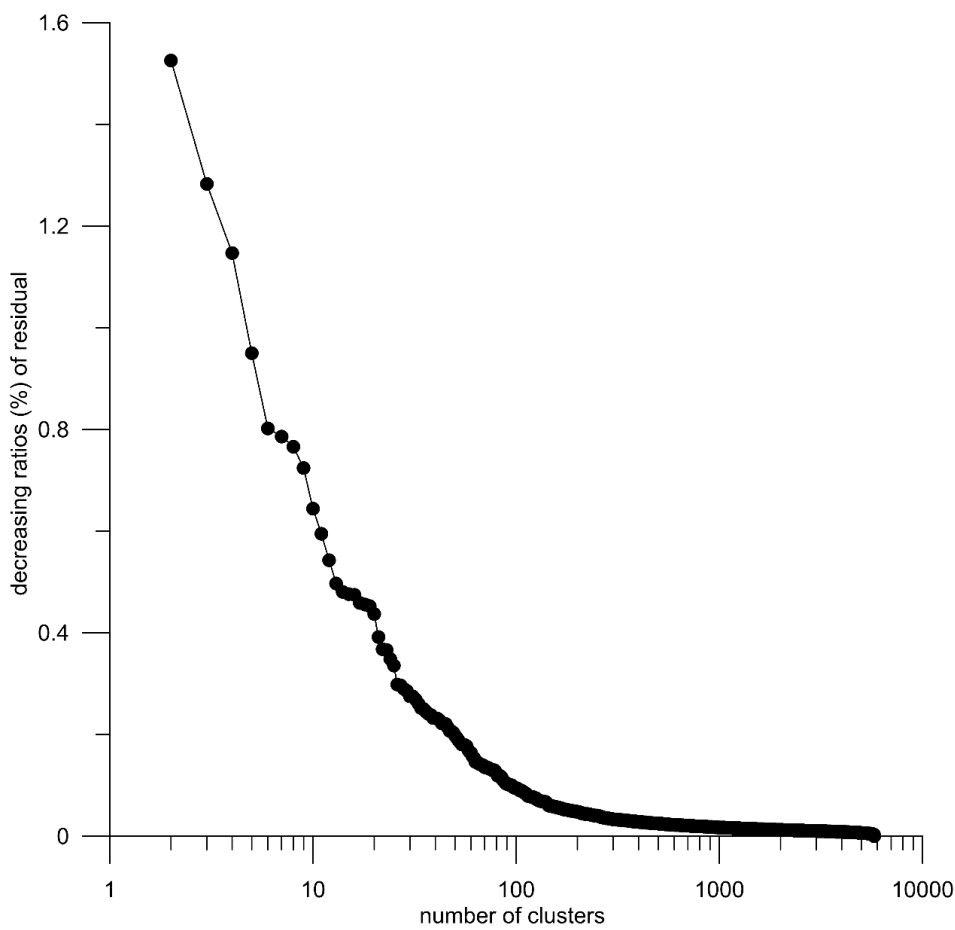

**Figure 2.** Relationship between number of clusters and decreasing ratios of residual.

The S1 topic (travel) included 389 (6.73%) articles; this was the topic category with the lowest proportion of articles. The top 10 words with high TF–IDF weights from most to least frequent were: *travel*, *destination*, *tourist*, *experience*, *behavior*, *model*, *social*, *relationship*, *market*, and *online*. The travel topic highly concerned destination, tourist, and experience, and several references [18,60] mentioned this topic.

The S2 topic (culture) included 927 (16.0%) articles; the top 10 words with high TF–IDF weights from most to least frequent were: *tourist*, *destination*, *experience*, *behavior*, *culture*, *model*, *image*, *travel*, *site*, and *social*. This topic covered multiple cultures, heritages, religions, and particular tastes, and several references [18,60] revealed this topic.

The S3 topic (sustainability) included 2153 (37.3%) articles; this was the topic category with the highest proportion of articles. The top 10 words with high TF–IDF weights from most to least frequent were: *development*, *social*, *community*, *leisure*, *culture*, *experience*, *local*, *sustainable*, *change*, and *nature*. The top 10 feature words were focused on sustainable development, social responsibility, social impact, community development, local development, local culture, or natural environment. This topic covered tourism sustainability, sustainability perceptions, sustainability management, impact on nature, social sustainability, and corporate social responsibility, and matched the topic of some academic researchers [20,61–63]. This topic also concerns issues related to community development and responsible tourism, sustainable development, resource management, sense of place and tourism experience, ecology, and biological well-being.

The S4 topic (model) included 666 (11.5%) articles, and the top 10 words with high TF–IDF weights from most to least frequent were: *model*, *economy*, *impact*, *effect*, *country*, *demand*, *policy*, *relationship*, *development*, and *tourist*. The top 10 feature words were highly correlated with the leisure model, economic model, regression model, economic impact or economic cost, demand, policies, and economic development. This topic involves regression model, logistic regression model, multiple regression model, evaluation model, economic model, correlation analysis, structural equation model, and theoretical model, and some references [64,65] demonstrated this topic.

The S5 topic (behavior) included 828 (14.3%) articles. The top 10 words with high TF–IDF weights from most to least frequent were: *behavior*, *service*, *intention*, *satisfaction*, *visitor*, *customer*, *experience*, *relationship*, *effect*, and *model*. The top 10 feature words are highly relevant to behavior, intention, service, and satisfaction, the experience of tourists or consumers, and also tourism quality. Most studies of this topic involved consumer behavior [66–68] and pro-environmental behavior in tourism [68–70].

The S6 topic (hotel) included 820 (14.2%) articles. The top 10 feature words with high TF–IDF weights from most to least frequent were: *destination*, *hotel*, *image*, *management*, *performance*, *effect*, *model*, *tourist*, *marketing*, and *strategy*. The top 10 feature vocabulary words are highly relevant to topics such as the location, image, performance, and management of hotels or activities, as well as marketing strategies related to tourism activities or hotels [18,60].

To evaluate the internal consistency of automatic topic classification within the same cluster, this study applied a cross-checking method to select the first three documents with the smallest Euclidean distances, and the titles, keywords, and Euclidean distances of these three articles of distinct topics are as follows (Table 4):

(1) Travel: The feature words of *visitor* and *travel* appeared in the title or keywords of these two documents (D3142 and D0032), and these two documents were highly related to this topic. Document D0114 presented the feature word *tourism* in the keywords and explored the female migrant laborers' employment experience and histories in the tourism industry, but was not highly related to this topic.

(2) Culture: Documents D1023 and D1902 investigated the culture-related activities of gastro-tourists and religion, respectively. Document D4497 explored destination image and utilized web 2.0 as a word-of-mouth communication tool. These contents of the first two documents fitted this topic.

(3)　Sustainability: The titles of these three documents (D2164, D2789, and D3037) mentioned social capital, citizenship behavior, and tourism–conservation enterprises, and these documents matched this topic.

**Table 4.** The titles and author keywords of the top three documents with the minimum Euclidean distances.

| Items | No. | Title | Author Keywords | Distance |
|---|---|---|---|---|
| S1 | D3142 | Visitors' engagement and authenticity: Japanese heritage consumption | Authenticity; Engagement; Japan; Heritage; Loyalty; Preconceived notions | 0.633 |
| | D0114 | Humanising migrant women's work | Women; Migration; Gender; Precarious work; One-voice research; Tourism labour | 0.639 |
| | D0036 | Vacation from work: A 'ticket to creativity'? The effects of recreational travel on cognitive flexibility and originality | Travel; Vacation; Holiday; Creativity; Flexibility; Originality; Innovation | 0.659 |
| S2 | D1023 | Attributes of Memorable Gastro-Tourists' Experiences | sustainable gastro-tourism development; memorability; co-creation; stakeholder theory; food or culinary tourism; destination branding | 0.600 |
| | D1902 | Understanding tourists in religious destinations: A social distance perspective | Social distance; Pilgrimage; Lumbini; Buddhists; Religious motives; Heritage tourism; Communitas | 0.606 |
| | D4497 | The new role of tourists in destination image formation | tourism image; image-formation process; Web 2.0; word-of-mouth; information and communications technologies | 0.608 |
| S3 | D2164 | Social capital and destination strategic planning | Tourism strategic planning; Social capital; Bonding social capital; Bridging social capital; Stakeholders; Cooperation; Trust; Reciprocity | 0.303 |
| | D2789 | Networks, citizenship behaviours and destination effectiveness: a comparative study of two Chinese rural tourism destinations | rural tourism; social network analysis; community citizenship behaviours; destination effectiveness; social capital; tourism operators | 0.867 |
| | D3037 | Tourism-conservation enterprises as a land-use strategy in Kenya | Africa; tourism; conservation enterprises; African Wildlife Foundation; Kenya; Koija Starbeds; institutional arrangements | 0.874 |
| S4 | D5515 | China's outward foreign direct investment in tourism | Outward foreign direct investment; Country choice; Tourism; China | 0.816 |
| | D1485 | Willingness to pay for flying carbon neutral in Australia: an exploratory study of offsetter profiles | voluntary carbon offsets; willingness to pay; discrete choice modelling; attitude-behaviour; offsetter profiles; climate change | 0.820 |
| | D1743 | Unplanned Tourist Attraction Visits by Travellers | Unplanned stops; trip plan; en route decision | 0.821 |
| S5 | D3389 | Predicting determinants of hotel success and development using Structural Equation Modelling (SEM)-ANFIS method | Hotel success and development; Tourism; Critical Success Factors (CSFs); TOE framework; HOT-fit Model; SEM-ANFIS | 0.598 |
| | D5582 | The effect of promotion on gaming revenue: A study of the US casino industry | Promotion; Gaming revenue; Interaction effect; Casino industry | 0.612 |

**Table 4.** *Cont.*

| Items | No. | Title | Author Keywords | Distance |
|---|---|---|---|---|
| | D1121 | Tourist districts and internationalization of hotel firms | Tourist districts; Location advantages; Internationalization; Hotel industry; Knowledge spillovers | 0.619 |
| S6 | D0792 | Tourism demand in Hong Kong: income, prices, and visa restrictions | SARS; tourism demand; visa restrictions; policy implementation; co-integration analysis; error correction model | 0.797 |
| | D2817 | An environment-adjusted dynamic efficiency analysis of international tourist hotels in Taiwan | four-stage approach; dynamic data envelopment analysis (DEA); slacks-based measure (SBM); Tobit regression; international tourist hotels | 0.813 |
| | D0586 | How power distance affects online hotel ratings: The positive moderating roles of hotel chain and reviewers' travel experience | Online rating; TripAdvisor; Hotel; Power distance; Hotel chain; Reviewer travel experience; Multidimensional rating | 0.817 |

Euclidean distance: the Euclidean distance of individual document to central of individual cluster.

(4) Model: The titles of these three documents (D5515, D1485, and D1743) mentioned models in the investment in tourism (establishment of a negative binomial regression model), willingness to pay, and unplanned tourist attraction, and these documents matched this topic.

(5) Behavior: The titles of documents D3389 and D5582 mentioned behavior models regarding determinants of hotel success and effect of promotion on gaming revenue. Document D1121 applied industrial district approach principles to identify tourist holiday districts situated along the Spanish coastline, but was not highly related to the topic of "Behavior".

(6) Hotel: The feature word *hotel* appeared in the title of documents D2817 and D0586. Document D0792, the minimum Euclidean distance, explored the tourism demand in Hong Kong and focused on income, prices, and visa restrictions problems, but existed a low relationship to the topic.

A cross-check analysis on the titles, abstracts, and topic categories of the eighteen documents across the six topics confirmed that four articles had low relevance to the feature words of the specific topics of Travel, Behavior, and Hotel. For the Travel topic, a portion of the document title for the fourth article is "Vacation from work"; however, one author included *travel* as a keyword, while the abstract mentions recreational *travel*. A portion of the title for the fifth article is "impact of human crowding versus spatial crowding on visitor satisfaction at a festival vacation"; one author included visitor satisfaction as a keyword, and the abstract mentions visitors. For the Culture topic, the abstract of the fourth article examined how people evaluate a culturally familiar country, this paper clearly matches this topic. For the Behavior topic, the titles of the fourth and fifth articles include "travel information search behavior" and "psychological distance", respectively; these two papers are clearly in line with this topic. For the Hotel topic, the titles of the fourth and fifth articles include "hotel engineering facilities" and "minimization of environmental footprint in hotel", so these two papers are deemed to match this topic. The analytical results suggest that most articles ranked with the first three minimum Euclidean distances matched the topic classification, although a few articles did not; many of the fourth and fifth articles also fit the theme. Therefore, the use of the cross-checking method would lead to a consistence with respect to the title, abstract, and keywords for most topics and articles. The assistance and opinions of domain experts can help to confirm the consistency of topic classifications and document contents.

*3.3. Marketing Strategy and Web Chart*

The feature words listed in Table 3 that are highly related to the 7P marketing strategies were identified by the domain experts and are described as follows: (1) promotion (*satisfaction*, *experience*, and *nature*); (2) place (*destination*, *local*, and *hotel*); (3) product (*hotel*, *destination*, *nature*, and *activity*); (4) people (*tourist*, *service*, and *visitor*); (5) physical evidence (*hotel* and *nature*); (6) process (*model* and *activity*); and (7) price (*economy* and *value*). According to the analytical results of academic journals in this study, the feature words in the tourism industry pay special attention to distinct 7P marketing strategies. For example, the most dominant feature words are satisfaction, experience, and nature for the promotion marketing strategy, which also represents that, when the tourism industry develops the marketing strategy considering promotion, this study would suggest that emphasis should be on customer's past satisfaction and demonstrating experience and natural scenery that the customer can enjoy in the tourism activity. Utilizing co-word analysis, the web chart (where thicker lines represent a high degree of co-occurrence) shows the importance levels and correlations of the feature words. If the feature words are not classified, the importance of individual feature words cannot be clearly identified. However, all the feature words form a complicated relationship even without classification, which makes it impossible to clearly distinguish the correlation among them. The web chart also presents the importance of and relationship between the feature words for different topic clusters, as well as the importance of and relationship with the 7P marketing strategy, which is highly related to the feature words. With the assistance of a web chart, the importance of feature keywords and the relevance levels among them can easily be identified, which assists in confirming linkages to research topics, activities, methods, and characteristics of tourism activities. The first three feature words (Figure 3a) with the highest co-word occurrence were *tourist*, *destination*, and *model*. For the feature word *tourist*, the top four feature words with the highest co-occurrence were *destination* (841), *experience* (589), *model* (546), and *behavior* (449), where the number in parentheses represents the number of documents with co-occurrence. The feature words with high co-occurrence in *tourist* were concerned with consumer behavior, tourism destination, models for evaluating tourism activities, and consumer experience. For the feature word *destination*, the top four feature words with the highest co-occurrence were *tourist* (841), *model* (425), *development* (376), and *implication* (365); this result shows that the co-occurrence with *destination* was more focused on tourist destinations, models for deciding on a destination, community development, economic development, sustainable development, and implications for destination. For the feature word *model*, the top four feature words with the highest co-occurrence were *tourist* (546), *relationship* (447), *destination* (425), and *effect* (391). This result presents that the co-occurrence with *model* focuses on consumer satisfaction models, behavior models, influence models, and impact models. Considering the specific feature word *market* (Figure 3b), the direct terms in marketing strategies, the top four feature words with the highest co-occurrence were *tourist* (274), *destination* (232), *travel* (165), and *model* (161); this result suggests that *market* has a high co-occurrence with tourism markets, travel markets, and models related to tourism marketing. The relevance of paired feature words can be seen in Figure 3b, but the relevance of three or more feature words cannot be identified.

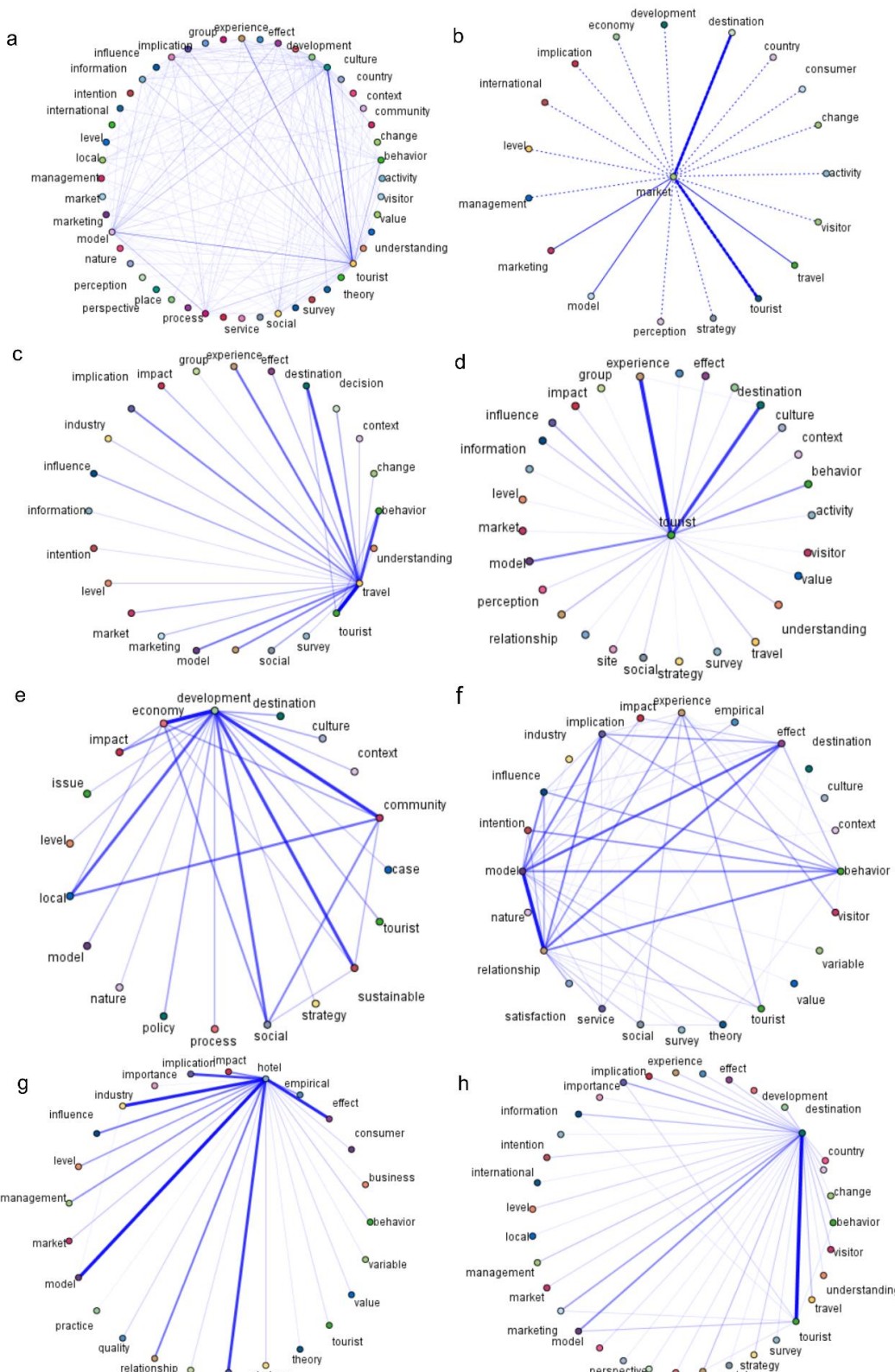

**Figure 3.** The web diagrams for (**a**) top seventy feature words; (**b**) feature word *market*; (**c**) Travel theme; (**d**) Culture theme; (**e**) Sustainability theme; (**f**) Model theme; (**g**) Behavior theme, and (**h**) Hotel theme.

Based on the classification of the aforementioned six topics, this study selected the first 70 feature words with high TF–IDF weights to perform the co-word analysis and visualize

the feature words that are highly related to the 7P marketing strategies. The results are as follows.

(1) The Travel theme showed co-occurrence across more than fifty articles (see Figure 3c), where the three feature words with high co-occurrence were (a) *travel*, *tourist*, and *destination*; and (b) *travel*, *tourist*, and *behavior*. The two feature words of high co-occurrence were (a) *travel* and *destination*, (b) *travel* and *behavior*, (c) *travel* and *experience*, (d) *travel* and *implication*, (e) *travel* and *tourist*, (f) *travel* and *model*, and (g) *travel* and *influence*. When considering marketing strategy, the Travel theme presented issues related to place, people, product, promotion, and process.

(2) The Culture theme showed co-occurrence across more than 100 articles (see Figure 3d), where Tourist appeared in the highest number of articles on this topic. The three feature words with high co-occurrence were (a) *tourist*, *experience*, and *destination*. Two feature words with high co-occurrence were (a) *tourist* and *destination*, (b) *tourist* and *behavior*, (c) *tourist* and *experience*, (d) *tourist* and *model*, (e) *tourist* and *culture*, and (f) *tourist* and *influence*. When considering marketing strategy, the Culture theme presented issues related to place, people, product, promotion, and process.

(3) The Sustainability theme showed co-occurrence across more than 140 articles (see Figure 3e). The feature words with high co-occurrence in this topic were diversified, including *sustainable*, *local*, *community*, *development*, *economic*, and *social*. The three characteristic words with high co-occurrence were (a) *local*, *community*, and *development*; (b) *sustainable*, *social*, and *development*; (c) *economic*, *community*, and *development*; and (d) *economic*, *community* and *social*. There were many two feature words with high co-occurrence, such as (a) *sustainable* and *development*, (b) *social* and *development*, (c) *community* and *development*, (d) *local* and *community*, (e) *local* and *development*. When considering marketing strategy, the Sustainability theme presented issues related to promotion, physical evidence, place, and product.

(4) The Model theme showed co-occurrence across more than 100 articles (see Figure 3f). The feature words with high co-occurrence in this topic were diversified, and included *model*, *intention*, *relationship*, *influence*, *implication*, *effect*, and *behavior*. The three feature words with high co-occurrence were (a) *model*, *intention*, and *behavior*; (b) *model*, *influence*, and *behavior*; (c) *model*, *implication*, and *behavior*; (d) *model*, *effect*, and *relationship*; (e) *experience*, *visitor* and *relationship*; (f) *economic*, *community*, and *social*; (g) *tourist*, *experience* and *relationship*; and (h) *intention*, *behavior*, and *relationship*. There were many two feature words with high co-occurrence, such as (a) *model* and *relationship*, (b) *model* and *intention*, (c) *model* and *influence*, (d) *model* and *effect*, (e) *model* and *behavior*. When considering marketing strategy, the model theme presented issues related to promotion, price, place, people, and product.

(5) The Behavior theme showed co-occurrence across more than 40 articles (see Figure 3g). The three feature words with high co-occurrence were *industry*, *model*, and *hotel*; there were many two feature words with high co-occurrence, such as (a) *effect* and *hotel*, (b) *hotel* and *service*, (c) *hotel* and *relationship*, (d) *hotel* and *model*, (e) *hotel* and *management*. When considering marketing strategy as related to the Behavior theme, all 7P marketing strategies were involved.

(6) The Hotel theme showed co-occurrence across more than 60 articles (see Figure 3h). The feature words with high co-occurrence in this theme focused on *destination* and *tourist*, which are also important management factors for the hotel industry. Three feature words with high co-occurrence were (a) *destination*, *tourist*, and *implication*; (b) *destination*, *tourist*, and *information*; (c) *destination*, *tourist*, and *marketing*; (d) *destination*, *tourist*, and *model*; and (e) *destination*, *tourist*, and *strategy*. There were many two feature words with high co-occurrence, such as (a) *destination* and *implication*, (b) *destination* and *information*, (c) *destination* and *marketing*, (d) *destination* and *model*, (e) *destination* and *tourist*. Considering marketing strategies within the Hotel theme, all 7P related issues were involved.

This paper utilized a web chart and co-word analysis to visualize the importance and interrelation of the feature words that were highly related to the 7P marketing strategies for the distinct topics. The web charts for the Hotel and Behavior themes demonstrated

the involvement of all 7P marketing strategies; however, the remaining four themes only included some of the marketing strategies.

*3.4. Strategy Diagram*

In strategy diagrams, the size of the area is proportional to the number of articles in the specific cluster (see Figure 4), where density measures the strength of relations in the same cluster, and centrality presents the connectivity levels to other clusters. According to descriptions made by Cobo et al. [71] on strategy diagrams, Zone I (motor theme) contains themes that are central to the construction of the tourism field; Zone II (basic and transversal themes) contains themes that are not central but already well-developed; Zone III (highly developed and isolated themes) contains themes that are both peripheral and little developed; and Zone IV (emerging or declining themes) contains themes that are uncentered and undeveloped, but that are becoming mature. The application of a strategic diagram provides information on connotations and trends for each theme as follows (Figure 4):

(1)     Culture (S2) and Behavior (S5), located in Zone I, represent the topic issues with the strongest maturity and cohesion in the tourism field, and they are at the center of the research issues; this means that Culture and Behavior have become complete and mature themes in the tourism field.

(2)     Travel (S1), located in Zone II, represents the theme issue that is highly interconnected but loosely cohesive, and this subregion contains basic, transversal, and generic subjects. The cohesion of feature words related to tourism topics still needs to be strengthened and deepened. In the past, most feature words on Travel topics focused on the application and validation of theories, demographic variables, implications or images, participating behaviors, tourist attitudes, tourist behaviors, degrees of involvement in tourism, past experiences, and socio-economic factors. To improve topic cohesion, new technology models, behavioral models, and theoretical models could be established or extended to deepen the relevant cases and research levels, such as through the use of big data analytics or artificial intelligence technology to condense and improve information technology related to the Travel theme.

(3)     The Hotel theme (S6), located in Zone III, represents the characteristics of the tourism field, which has low thematic interconnectivity but strong cohesion, and this theme is well-developed internally. Perhaps the Hotel theme is a relatively independent research theme, so the development of related concepts still needs to be strengthened, as well as establishment of connections with other themes. To improve the cohesion of this topic, new technology models, behavioral models, or theoretical models could be established or extended to deepen the relevant cases and research levels, such as through the use of big data analytics or artificial intelligence technology to condense and improve information technology related to the Hotel theme. Many research methods could be viable ways of expanding the scope of this subject, such as the utilization of theory and models for consumer behavior and various models to evaluate consumer behavior, hotel management performance, weights of driving factors on hotel service and satisfaction levels, green hotel management performance levels, green food satisfaction levels, and management performance under different types of organizational culture and leadership.

(4)     The themes of Sustainability (S3) and Model (S4), located in region IV, represent that both the external connection degree and internal cohesion of these themes are low: these themes have higher degrees of divergence and freedom and are not well-developed. For example, the Sustainability topic involves the environment, ecology, corporate social responsibility, corporate sustainability, sustainable development goals, sustainability policies, sustainable development, disaster prevention, natural resistance; the Model theme includes multiple regression models, logistic models, behavior models, economics models, financial models, and satisfaction models, among others.

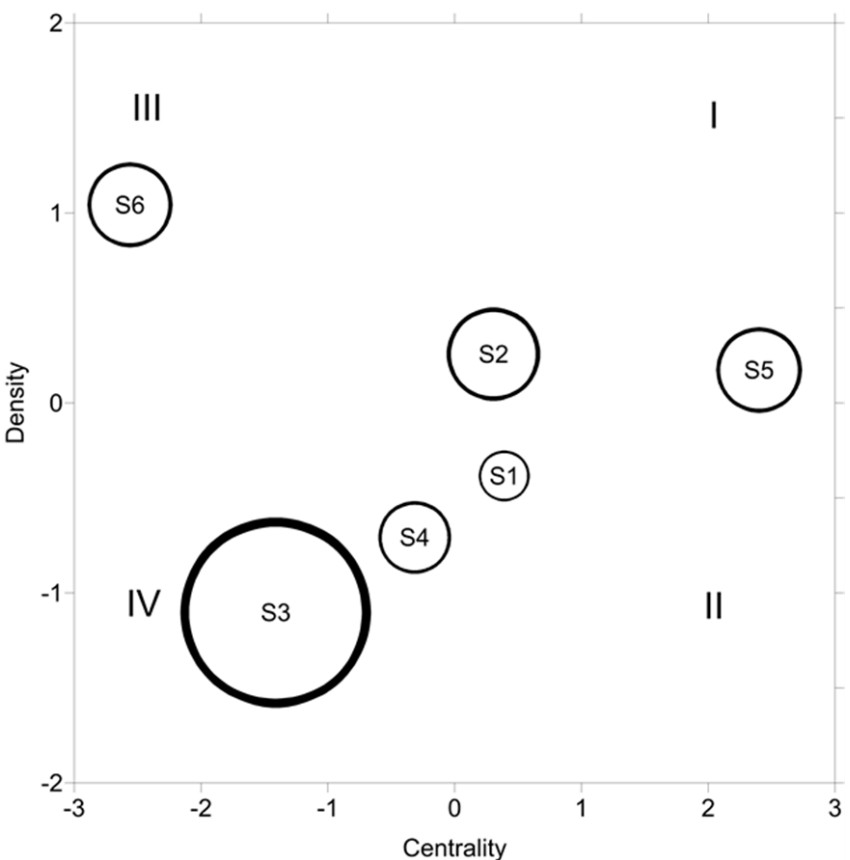

**Figure 4.** The strategy diagram.

In summary, the Culture and Behavior topics are found to be at the core of tourism field, in that they have been well-developed. The Travel topic has a great potential for development. The Hotel topic is a highly independent research theme with a high degree of internal cohesion, but its connection with other topics needs to be strengthened. The Sustainability and Model topics still require continuous research to enhance the internal convergence and external connectivity, as well as strengthen or highlight the relevant aspects in the tourism field.

## 4. Conclusions

This research utilized the text mining technique to automatically classify documents in the tourism field and screened academic journal articles from the WoS database for the years 2010–2019. This procedure involved applying text mining technology to segment the article abstracts, inviting domain experts in the leisure field to identify feature keywords and build a TF–IDF weights matrix, classifying themes and articles into six categories using a hierarchical K-means cluster analysis, employing a co-word analysis to analyze the characteristics and appropriateness of the various topics, using a cross-check analysis to confirm the consistency of the article classifications, and applying a strategy diagram to display the degrees of maturity and cohesion for the distinct themes and different topics. All of the above was undertaken to conduct an exploratory analysis of topic classification in the tourism industry and to provide qualitative and quantitative results. The various professional journals all have their own specific characteristics and development paths; over time, technological and environmental evolution may lead to changes in what they choose to focus on. Therefore, the establishment of a standard operation process can assist in automatic document classification.

This study provided six main contributions: (1) established a reasonable and feasible automatic classification process for tourism articles and verified the correctness of

the topic classification; (2) used a cross-check manner to confirm the consistency of the internal classification of the articles in each category; (3) identified the feature words of the six themes in the tourism field; (4) used a web diagram to clearly distinguish the characteristics, importance, relevance, and differences of feature words across the six theme categories; (5) introduced the 7P marketing strategy into the co-word analysis and made connections between marketing strategy and feature words applicable to the different theme categories; and (6) emphasized the importance of domain expert participation in the text mining process at this stage with respect to determining feature words in word segmentation, deciding on the number of topic categories and comparing the consistency of articles.

Automatic document classification has undergone major changes in recent years due to the invention of powerful new tools; this area is worthy of further investigation, especially in the face of current and future applications of big data and artificial intelligence. Currently, text-mining still requires the participation of domain experts to ensure the consistency, correctness, and recognition of professional vocabulary and the identification of synonyms. A good dictionary or knowledge base for domain terms will assist future researchers in automatic document classification. The combination of web-chart and topic classification could provide precise feature words with high co-occurrence for distinct topics and increase the understanding and application of feature words. Many tourism industries have websites that could collect the evaluation data of customers engaged in tourism activities. Text mining could also analyze and classify the positive and negative feedback opinions from customers and provide different marketing strategies and specific concessions based on the feedback opinions. In addition to the qualitative statements of text mining, the combination of quantitative data from feedback opinions would provide tourism industries with more precise and customized marketing strategies.

**Author Contributions:** Conceptualization, J.-S.H., C.-H.L., S.-F.C. and T.-Y.Y.; methodology, I.-C.C. and T.-Y.Y.; software, I.-C.C. and T.-Y.Y.; validation, J.-S.H., C.-H.L., S.-F.C., T.-Y.Y. and I.-C.C.; formal analysis, I.-C.C. and T.-Y.Y.; investigation, J.-S.H., I.-C.C. and T.-Y.Y.; resources, J.-S.H., C.-H.L., S.-F.C. and T.-Y.Y.; data curation, I.-C.C. and T.-Y.Y.; writing—original draft preparation, T.-Y.Y.; writing—review and editing, J.-S.H., C.-H.L. and S.-F.C.; visualization, I.-C.C. and T.-Y.Y.; supervision, J.-S.H.; project administration, J.-S.H.; funding acquisition, J.-S.H. and T.-Y.Y. All authors have read and agreed to the published version of the manuscript.

**Funding:** The authors express their gratitude to the Ministry of Science and Technology, Taiwan (MOST 109-2511-H-130-003-MY2) for funding this project.

**Institutional Review Board Statement:** Not applicable.

**Informed Consent Statement:** Not applicable.

**Data Availability Statement:** Not applicable.

**Conflicts of Interest:** The authors declare no conflict of interest.

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
