# Peer review of "Exploration of Topic Classification in the Tourism Field with Text Mining Technology—A Case Study of the Academic Journal Papers"

_sustainability, doi:10.3390/su14074053_

Round 1

Reviewer 1 Report

The article is dedicated to a contemporary, interesting and important topic. It shows both the application of natural language processing technologies and the importance of experts in a given field, which are crucial for improving analysis.

The introduction is extensive and effectively provides an overview of the literature on the subject. My recommendation is that the introduction should include an aim and research hypotheses, and the review of the literature should be a separate, second point.

The point Materials and Methods could be expanded and the analysis approach should be presented schematically, or through a diagram. The concluisons and summaries int he Results and Discussion point, are well developed and show in practice how the analysis was done.

Author Response

Q1

The article is dedicated to a contemporary, interesting and important topic. It shows both the application of natural language processing technologies and the importance of experts in a given field, which are crucial for improving analysis.

A1

Thank you.

Q2

The introduction is extensive and effectively provides an overview of the literature on the subject. My recommendation is that the introduction should include an aim and research hypotheses, and the review of the literature should be a separate, second point.

A2

Thank you for the comment.

Revised on lines 55-72. We rearrange the sentences in these two sections.

Lines 55-66.

Information technology is part of the lifeblood of the tourism industry [7]. Combining knowledge gathered through statistics and domain experts from the tourism industry can help verify the results of visualization analysis. Information technology can apply an automatic topic classification to natural language processing documents to classify representative documents quickly and objectively; co-word analysis and association rule analysis can then be used to analyze the importance and relevance of specific words. There are four main research aims for this article: (1) carry out the subject classification process of academic articles in the tourism field to assess the consistency and characteristics of the topic classification; (2) assess the characteristics of the subject classification and confirm its consistency; and (3) use co-word analysis and strategic diagram to understand the importance and relevance of specific marketing strategy vocabulary; and (4) recognize the research tendencies of distinct topics in tourism field.

Lines 67-72

In the past, the process of classifying, deconstructing, and analyzing relevant documents in this area required significant time and resources from domain experts. As this body or work continues to grow and diversify, text mining technology can better comprehend and promote the leisure industry, such that the general public be-comes willing to understand, recognize, support, and participate in achieving the goals of sustainable development.

Q3

The point Materials and Methods could be expanded and the analysis approach should be presented schematically, or through a diagram. The concluisons and summaries in the Results and Discussion point, are well developed and show in practice how the analysis was done.

A3

Thank you for the comment.

Revised on lines 237-241.

The detailed five steps for extracting feature words were shown as (1) preprocessed text cleaning for abstracts of journals papers; (2) word segmentation with Jieba; (3) filter tokens, filter stopped words, perform other cleaning rules and replace tokens; (4) construct and confirm of feature wordlist with domain experts, and (5) calculate TF-IDF weights.

Reviewer 2 Report

An interesting article and one adding to the discourse around textual analysis as a means for discovering connections and trends in the acad tourism literature.

I wonder though, in this and similar articles, if such an approach is, yes,  creating an internally consistent world view of say tourism topics and elements, through classifications and textual analysis but not yet demonstrating how this world view connects to and can benefit business and other stakeholders in the real world (as is suggested in your article - lines 35 - 38).

The verification process described seems to be again taken within the articles world view, i.e., by 'experts' but he process seems to be focused on internal consistency of the model and results, rather than linking the results to the 'real world'.

However, many useful insights and an effective exposition of the current state of textual analysis and a very useful demonstration of the strengths (and perhaps current limitations) of the approach.

Notwithstanding the above, the results (s 4.4) raise some interesting ideas and insights around key topics in current tourism industry and research.

I wonder if a new 'areas for further research' section could take these insights and suggest where to go next in terms of further investigating their meaning for tourism academia and practitioners.  Such a new section could perhaps also canvas some of the issues raised above and suggest means for real-world validation of the usefulness of the techniques advocated here. 

Some specific suggestions:

Add a source for your definition of tourism (line 41)

'...text mining technology is required..': certainly text mining can assist, but not sure the case is yet made strongly enough to 'require' such an approach for success (line 56)

This technique can identify (= suggest?) '... emerging research topics, methods...' as Okumas found, but this is not the same as using this technique as a normative approach to industry analysis and as a basis for industry development (as suggested in lines 78 - 86)

This is also suggested at lines 124 - 130 - there are many research themes and influencing factors (lines 124-125), and the interp ability of the local industry can increase the income of sust tourism (lines 126-39), but not sure the article sufficiently makes the case for this approach providing firm enough guidance for the industry to act on.

Interesting discussion on marketing (section 1.2) but can't quite see where you derive and define your 7 P model which then reappears at line 450 ff as seemingly defined and proven.

Maybe re-work the language around sustainability and marketing ( Sustainability is a BRIGHT marketing strategy' line 148)

Linked to me comments above, I'd be more cautious about claiming '...tourism management is inseparable from information technology..' (line 165).  I think for large well-resourced organisations this may be so, but for smaller organisations, or those in less developed markets and with a wide range of consumers who may not be interested in providing the data required for sophisticated data collection and analysis, this my not be (yet) so (lines 165-70).

Author Response

Reviewer #2

Q1

An interesting article and one adding to the discourse around textual analysis as a means for discovering connections and trends in the acad tourism literature.

A1

Thank you.

Q2

I wonder though, in this and similar articles, if such an approach is, yes, creating an internally consistent world view of say tourism topics and elements, through classifications and textual analysis but not yet demonstrating how this world view connects to and can benefit business and other stakeholders in the real world (as is suggested in your article - lines 35 - 38).

A2

Thank you for this comment.

This paper selected 8 SSCI journals (5783 papers) in the subject field of hospitality, leisure, sport and tourism from the WoS database and analyze the abstracts using a data driven manner. Therefore, this paper applies text mining technique, K-means clustering manner and assistance of domain experts to determine the numbers of topic classification and internally consistence within the same cluster. A systematic classification is a crucial issue related to several principals for any specific domain. The web-chart of distinct cluster presents the feature words with high co-occurrence, and also demonstrated the important issues that academic papers considered.      

Q3

The verification process described seems to be again taken within the articles world view, i.e., by 'experts' but he process seems to be focused on internal consistency of the model and results, rather than linking the results to the 'real world'.

A3

Thank you for this comment. The “experts” terms were revised to domain experts. 

Revised on lines 241-245.

Three domain experts with domain knowledge in the tourism field were invited to extract, confirm, and determine feature keywords, calculate TF-IDF weights, build the DTM matrix, perform both topic classification and co-word analysis, verify text mining results, and extract the required information, as well as verify the consistency of the topic classification and corpus data.

Q4

However, many useful insights and an effective exposition of the current state of textual analysis and a very useful demonstration of the strengths (and perhaps current limitations) of the approach.

A4

Thank you.

Q5

Notwithstanding the above, the results (s 4.4) raise some interesting ideas and insights around key topics in current tourism industry and research.

A5

Thank you

Q6

I wonder if a new 'areas for further research' section could take these insights and suggest where to go next in terms of further investigating their meaning for tourism academia and practitioners. Such a new section could perhaps also canvas some of the issues raised above and suggest means for real-world validation of the usefulness of the techniques advocated here.

A6

Thank you for this comment.

Revised on lines 645-653.

The combination of web-chart and topic classification could provide precise feature words with high co-occurrence for distinct topics, and increase the understanding and application of feature words. Many tourism industries have websites that could collect the evaluation data of customers engaged in tourism activities. Text mining could also analyze and classify the positive and negative feedback opinions from customers, and provide different marketing strategies and specific concessions based on the feedback opinions. In addition to the qualitative statements of text mining, the combination of quantitative data from feedback opinions would provide tourism industries with more precise and customized marketing strategies.

Q7

Add a source for your definition of tourism (line 41)

A7

Thank you for this comment.

The reference of definition is cited as reference [3].

3.Björk, P. Definition paradoxes: from concept to definition. Critical issues in ecotourism: Understanding a complex tourism phenomenon. 2007, 23-45.

Q8

'...text mining technology is required..': certainly text mining can assist, but not sure the case is yet made strongly enough to 'require' such an approach for success (line 56)

A8

Thank you for this comment.

Revised on lines 68-72.

As this body or work continues to grow and diversify, text mining technology can better comprehend and promote the leisure industry, such that the general public be-comes willing to understand, recognize, support, and participate in achieving the goals of sustainable development.

Q9

This technique can identify (= suggest?) '... emerging research topics, methods...' as Okumas found, but this is not the same as using this technique as a normative approach to industry analysis and as a basis for industry development (as suggested in lines 78 - 86)

A9

Thank you for this comment. Revised on lines 79-81. The original sentences were revised as following:

and identified emerging research topics, methods, and areas of national or interdisciplinary cooperation. Most of the 462 articles centered on gourmet, quantitative, and practical topics.

Q10

This is also suggested at lines 124 - 130 - there are many research themes and influencing factors (lines 124-125), and the interp ability of the local industry can increase the income of sust tourism (lines 126-39), but not sure the article sufficiently makes the case for this approach providing firm enough guidance for the industry to act on.

A10

Thank you for this comment. Revised on lines 124-127. The original sentences were revised as following:

The interpretation ability of the local tourism industry could increase the income of sustainable tourism, and local interpreters could meet customer needs and create local employment, promote economic sustainability, and also act as on-site supervisors of visitors to influence their understanding of local perspectives…

Q11

Interesting discussion on marketing (section 1.2) but can't quite see where you derive and define your 7 P model which then reappears at line 450 ff as seemingly defined and proven.

A11

Thank you for this comment. Revised in lines 454-460. We add the following sentences.

According to the analytical results of academic journals in this study, the feature words in the tourism industry pays special attention to distinct 7p marketing strategies. For example, the most dominant feature words are satisfaction, experience, and nature for the promotion marketing strategy, which also represents that when the tourism industry develops the marketing strategy considering about promotion, this study would suggest that emphasize customer's past satisfaction, demonstrate experience and natural scenery that customer can enjoy in the tourism activity.

Q12

Maybe re-work the language around sustainability and marketing (Sustainability is a BRIGHT marketing strategy' line 148)

A12

Thank you for this comment.

The original sentence was deleted.

Q13

Linked to me comments above, I'd be more cautious about claiming '...tourism management is inseparable from information technology.' (line 165). I think for large well-resourced organisations this may be so, but for smaller organisations, or those in less developed markets and with a wide range of consumers who may not be interested in providing the data required for sophisticated data collection and analysis, this my not be (yet) so (lines 165-70).

A13

Thank you for this comment. Revised on lines 161-163. The original sentence was revised as following:

Based on the existing information technology, the tourism industry can expand consumerism-based IT and tourism research in order to participate in a wider dialogue; …